# OmniBal: Towards Fast Instruct-tuning for Vision-Language Models via Omniverse Computation Balance

## Abstract

Vision-language instruct-tuning models have recently made significant progress due to their more comprehensive understanding of the world. In this work, we discover that large-scale 3D parallel training on those models leads to an imbalanced computation load across different devices. The vision and language parts are inherently heterogeneous: their data distribution and model architecture differ significantly, which affects distributed training efficiency. To address this issue, we rebalance the computational load from data, model, and memory perspectives, achieving more balanced computation across devices. Specifically, for the data, instances are grouped into new balanced mini-batches within and across devices. A search-based method is employed for the model to achieve a more balanced partitioning. For memory optimization, we adaptively adjust the re-computation strategy for each partition to utilize the available memory fully. These three perspectives are not independent but are closely connected, forming an omniverse balanced training framework. extensive experiments are conducted to validate the effectiveness of our method. Compared with the open-source training code of InternVL-Chat, training time is reduced greatly, achieving about 1.8x speed-up. Our method's efficacy and generalizability are further validated across various models and datasets. Codes will be released at https://github.com/anonymousiclr293/omnibal_example.

## 1 Introduction

Large language models (LLM) have brought new possibilities to many fields. Multi-modal models, particularly Vision-Language Models (VLMs) Alayrac et al. (2022); Team et al. (2023a); Reid et al. (2024); Liu et al. (2023a); Bai et al. (2023b); Chen et al. (2023), are advancing rapidly due to their deeper understanding of the world. The training scale of Vision-Language Models (VLMs) continues to expand, with increasingly larger datasets incorporating more text and higher-resolution images. Compared with the LLaVA-1.5 Liu et al. (2023a), the InternVL-Chat Chen et al. (2024) has expanded the dataset size from 665K to 5M and increased image resolution from 336x336 to 3840x2160. At the model level, larger vision encoders are adopted. The InternVL-Chat upgrades the visual encoder from ∼300M ViT-L-336px Radford et al. (2021) to ∼6B InternViT-448px Chen et al. (2023). The larger datasets and models result in a more time-consuming training process. Therefore, efficient training strategies are essential for the rapid advancement of the field.

3D parallelism Shoeybi et al. (2019); Rajbhandari et al. (2020); microsoft (2020) is a popular framework for large-scale distributed training, which allows data and models to be distributed across multiple devices. Balancing computational load across devices is crucial in 3D parallelism by minimizing idle times.

In this work, we find that for instruct-tuning large vision-language models, the heterogeneous nature of data and model structures brings new challenges to 3D parallelism training: (1) Varying input sizes of LLM and VIT cause imbalance computational loads across training iterations and devices. (2) The heterogeneity between LLM and VIT models leads to inherent differences in the computational load of their transformer blocks. Along with varying input sizes, this inevitably results in uneven computational load and computational bubbles. (3) Input size variation and computational

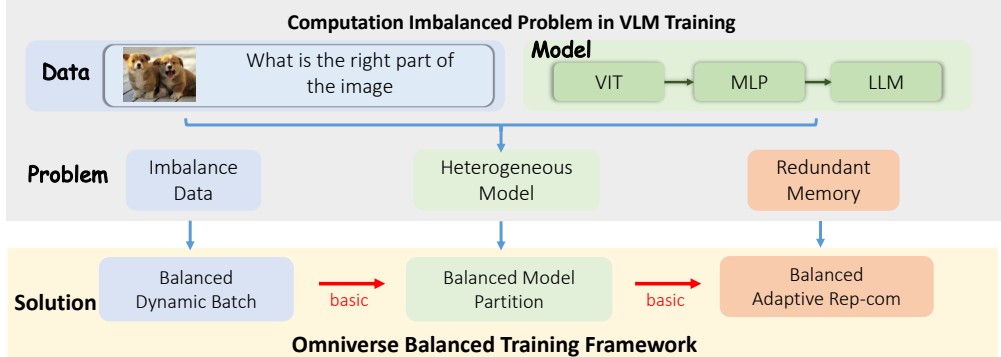

Figure 1: Overview of the computation imbalanced problem and our proposed solution in Standard Vision-Language instruct-tuning framework. We consider the bottleneck issues of data, model, and memory, and propose an omniverse solution addressing these three aspects, each providing the foundation for the next.

imbalance compel us to use the most aggressive re-computation (checkpointing) Li et al. (2014) strategy to prevent program crashes, which wastes computational resources. We refer to those issues caused by the heterogeneity in data and model structures in large vision-language models as the **Computation Imbalance** problem, which reduces training efficiency.

To address this problem, a simple and efficient training framework called Omniverse Balance (**OmniBal**) is proposed, to balance computational load across multiple devices. This framework systematically balances computation in three bottlenecks, *i.e.* data, model, and memory, as shown in Figure 1. OmniBal works in these three closely connected aspects. Data lays the groundwork for addressing model imbalances, while data and model form the foundation for solving memory issues. Ultimately, these three aspects collaborate to achieve balanced computation. **Data:** The balanced dynamic mini-batch method is proposed to group instances as new mini-batches according to text length and number of images. Specifically, an iterative algorithm based on sampling and filtering combines data of different sizes into balanced groups, ensuring stable input sizes; **Model:** We propose balanced model partitioning to evenly spread the computational load of LLM and VIT across devices. Using a search-based approach, we efficiently find optimal partition strategies within a small search space, enabling adaptation to different model architectures and hardware platforms. The balanced dynamic mini-batch method facilitates balanced model partitioning by ensuring input sizes are consistent in advance. **Memory:** A balanced adaptive re-computation method is proposed to optimize the re-computation strategy on each device, maximizing both memory utilization and training speed. We calculate the memory requirements of different models to adjust the re-computation strategy adaptively. Notably, our proposed balanced dynamic mini-batch and model partitioning ensure balanced computational loads on each device, making memory analysis feasible.

Extensive experiments are performed on various open-source VLM models at different scales, reducing overall training times significantly. GPU days are reduced for InternVL-Chat-1.5 (6+20B) from 61.8 to 21.3 under the Megatron-DeepSpeed microsoft (2020) backend. Scaling up to InternVL-Chat-1.5-Plus (6+34B), we consistently observe great speed-up, from 75.4 to 30.5 GPU days. We conduct thorough generalization experiments, including various datasets, hardware configurations, and multiple model combinations. Consistent and substantial improvements are observed across all experiments, demonstrating the effectiveness and versatility of our method.

## 2 RELATED WORK

### 2.1 MULTI-MODAL LARGE LANGUAGE MODEL(MLLM)

Large language models, such as ChatGPT OpenAI (2023a), GPT-4 OpenAI (2023b), Llama series Touvron et al. (2023a;b); AI (2024), and Gemini series Team et al. (2023b); Reid et al. (2024), have seen significant advancements recently. They rely on large datasets for training to achieve strong performance, particularly in few-shot and zero-shot scenarios. Typically, they are built on textual

Table 1: Analysis of computation imbalance. Time and Memory represent forward time and cost memory. t indicates token, and STD stands for standard deviation.

| Imbalance Dimension | Input Mean $\pm$ STD (t) | Time Mean $\pm$ STD (ms) | Memory Mean $\pm$ STD (G) |
|---|---|---|---|
| Inter-Stage | $1420 \pm 955$ | $85 \pm 93$ | $39 \pm 23$ |
| Intra-Stage-1 | $1975 \pm 1272$ | $136 \pm 155$ | $73 \pm 6$ |

data and can only accept text inputs. However, real-world scenarios often involve rich multi-modal information, *e.g.*, images. It has driven the development of large vision language models (VLMs). Visual encoders like Vision Transformer (ViT) Dosovitskiy et al. (2021) usually incorporate vision information. A cross-modal connector is also required to align the vision encoder outputs to the language models. LLaVA Touvron et al. (2023a) uses the simplest MLP, BLIP series Li et al. (2022; 2023); Dai et al. (2024) uses Q-former, Qwen-VL-Chat Bai et al. (2023b) uses a cross-attention module. VLMs expand large language models' capabilities and application scenarios by instruct-tuning with text and image data. However, introducing multi-modal data and heterogeneous encoders also brings challenges to the model training.

## 2.2 LARGE-SCALE DISTRIBUTED TRAINING

Distributed training is essential for efficiently utilizing multiple GPUs to train large language models. It is achieved through 3D parallelism Shoeybi et al. (2019); Rajbhandari et al. (2020); microsoft (2020): data, tensor, and pipeline parallelism. *Data Parallelism* splits the entire dataset into mini-batches and assigns them to multiple devices, each with a model replica. This approach maximizes the use of GPU power for large datasets. DeepSpeed Zero Rajbhandari et al. (2020) enhances it by reducing weight redundancy. However, it can still be challenged by the memory limits of individual devices when handling huge models. *Tensor Parallelism* distributes a model's weight matrices across multiple devices, enabling parallel matrix operations Shoeybi et al. (2019) and reducing per-device memory requirements. This method accelerates computation but requires dense inter-device communication, typically restricted to single-node deployments to minimize latency. *Pipeline Parallelism* divides a model into segments and assigns them to different devices, creating a computation flow like a production line. This technique facilitates larger model scaling across nodes. GPipe Huang et al. (2019) proposes micro-batching to decrease forward bubbles. PipeDream Narayanan et al. (2019) further proposes a one-forward-one-backward (1F1B) scheme to optimize memory usage. In pipeline parallelism, uneven layer partitioning can cause significant pipeline bubbles. PipeDream Narayanan et al. (2019) and AdaPipe Sun et al. (2024) optimize model partitioning and re-computation strategies based on profiling and dynamic programming, respectively. However, these advancements are primarily tested in text-based models and may require adaptation for large vision language model scenarios.

## 3 COMPUTATION IMBALANCE

In this section, we explore the unique challenges of large-scale distributed training for vision-language models, focusing on two dimensions: **Inter-Stage** and **Intra-Stage** computation imbalance. Inter-Stage means the computation imbalance of different pipeline parallel stages. Intra-Stage indicates the computation imbalance of the same stage across time and device. Figure 2 shows these two computation imbalances more intuitively. And they both include three specific levels: data, model, and memory. To quantify this problem, we used the InternVL-Chat-1.2 dataset Chen et al. (2024) to perform profile statistics shown in Tabel 1. For the Intra-Stage, we counted the information of Stage 1 as a sample.

**Data Imbalance:** LLMs are trained on texts using next-token prediction, allowing consistent input lengths through arbitrary text sub-strings. In contrast, VLMs handle texts and images, requiring data integrity, and preventing arbitrary truncation. The varying number of images, resolutions, and text lengths result in considerable differences in input sizes across mini-batches. From Tabel 1 and Figure 2, data imbalance occurs in Inter-Stage and Intra-Stage. To better quantify the impact of dynamic input, we define the DistRatio (introduced in Section 4) to measure the degree of data imbalance of VIT and LLM.

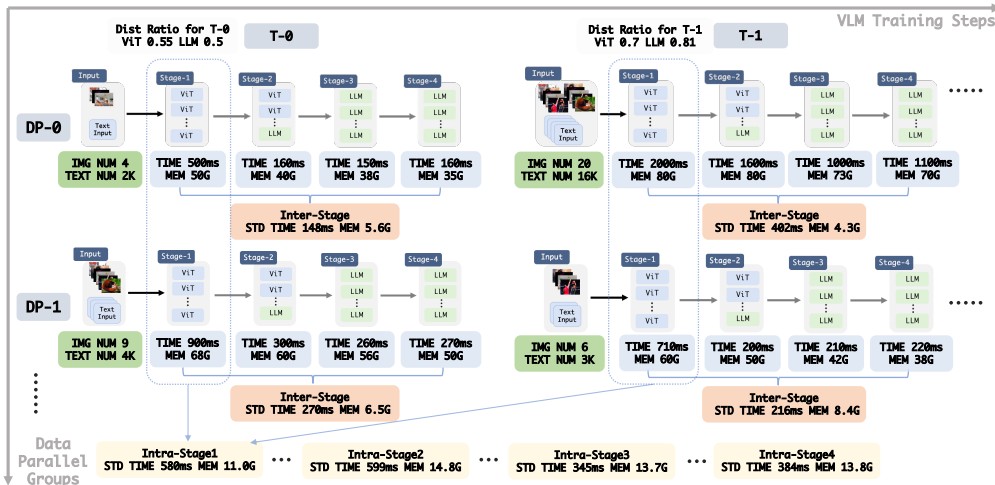

Figure 2: The Problem of Computation Imbalance in VLM Instruct-tuning Training Pipeline. DP-0 and DP-1 represent different Data Parallel processes. T-0 and T-1 represent different training times. TIME and MEM represent forward time and cost memory in the current stage respectively. STD stands for standard deviation.

**Model Imbalance:** LLMs use identical transformer modules with the same computational load. Evenly dividing these layers in pipeline parallelism distributes the load effectively. However, VLMs require additional image pre-processing, necessitating an image encoder. The structural disparity between VIT and LLM results in different computational demands. As shown by Tabel 1 and Figure 2, the standard deviation of forward time is huge in both Inter-Stage and Intra-Stage, indicating a serious computation imbalance.

**Memory Imbalance:** LLMs require significant GPU memory due to their large parameter size. When memory is insufficient, re-computation Li et al. (2014) techniques discard some intermediate activation values and recompute them during backward propagation to save memory. VLM encounters great memory challenges due to the variable scales of data inputs and the heterogeneity between vision and language models. The presence of numerous images or long text inputs can lead to excessive GPU memory usage, requiring the most aggressive re-computation settings to prevent the program from crashing. However, excessive re-computation can slow down the training process. From Table 1 and Figure 2, under the existing training setting, memory imbalance is reflected in both Inter-Stage and Intra-Stage.

**Differences between VLM and LLM training**: As mentioned above, the difference between VLM and LLM arises from the data composition and model structure, resulting in unique Inter-Stage and Intra-Stage challenges. Inter-Stage: Since LLM has a fixed structure, the model can be equally divided and there is no Inter-Stage imbalance for any input. Dynamic input or inconsistent text-image ratios in heterogeneous VLM will lead to an Inter-Stage imbalance problem. Intra-Stage: For the LLM-Pretrain task, the input is fixed and there is no Intra-Stage imbalance problem. Dynamic input can be converted into static input by simple packing Kosec et al. (2021) to reduce the computation imbalance for the LLM-SFT task. However, VLM instruct-tuning training cannot rely on simple packing to ensure fixed inputs for VIT and LLM, resulting in computation imbalance problems.

## 4 METHOD

This section presents our computation-balanced framework OmniBal for training large vision-language models. To address an imbalanced computational load across devices, we first manage the large data input variations, which is the most fundamental issue in the computation imbalance problem. This enables the possibility of balanced model partitioning. Finally, the re-computation strategy for each partition is optimized. Appendix A.1.2 shows our training pipeline.

## 4.1 BALANCED DYNAMIC MINI-BATCH

For instruct-tuning VLMs, each training sample contains various images and texts, resulting in non-fixed input sizes. We evaluate data imbalance from two perspectives: within-device samples and cross-device mini-batches.

**Pad Ratio (within-device):** When combining samples of different sizes into a mini-batch, smaller samples need to be padded to ensure aligned input sizes. The Pad Ratio is calculated as follows:

$$PadRatio = \frac{\sum_i^B (t_{max} - t_i)}{t_{max} \times B} \tag{1}$$

Where $t_{max}$ represents the maximum number of tokens in a mini-batch of size $B$, and $t_i$ denotes the number of tokens for sample $i$ within that mini-batch.

**Dist ratio (cross-device):** Even after padding, the sizes of mini-batches on different devices may vary, leading to different input scales across devices. The distribution ratio is calculated as follows:

$$DistRaito = \frac{\sum_i^N (T_{max} - T_i)}{T_{max} \times N} \tag{2}$$

Where $N$ represents the number of devices, $T_{max}$ denotes the maximum number of mini-batch tokens across all devices, and $T_i$ refers to the number of mini-batch tokens on the $i^{th}$ device. Non-fixed input sizes in VLMs have a larger Pad Ratio and Dist Ratio, as shown in Table 5 (row 1). A high Pad Ratio wastes computational resources, while a high Dist Ratio causes device idle time. They significantly impact training throughput efficiency.

To address this issue, An adaptive grouping strategy that organizes multiple samples, ensuring that both image and text sizes in the resulting groups remain within a relatively fixed range is implemented. We refer to this method as the Balanced Dynamic Mini-Batch. Determining the optimal grouping strategy is a non-trivial problem, An iterative method is designed using sampling and filtering to group samples. As illustrated in Algorithm 1 and Algorithm 2, our method **Iterative Sampling and Filtering (ISF)** involves the following steps:

*1.Sampling Stage:* For current dataset $\mathcal{D} = \{(x_i, y_i) \mid i\}$, we randomly add samples $d_i$ consisted of images $x_i$, text $y_i$ to current group $\mathcal{G}$. If the total number of images $I_v = \sum_{x_i \in \mathcal{G}} |x_i|$ or the total text length $I_t = \sum_{y_i \in \mathcal{G}} |y_i|$ reaches the predefined maximum number of images $Q_v$ or text $Q_t$, we add this group to the candidate set $\mathcal{P}$ and create a new group containing $(x_i, y_i)$ for the subsequent samples. Otherwise, we will continue adding samples to the current group. At the end of the sampling stage, we will have a candidate set $\mathcal{P} = \{\mathcal{G}_i | i = 1, 2, 3..\}$.

*2.Filtering Stage:* We first define the target number of images $Q'_v$ and text $Q'_t$. For each group $\mathcal{G}_i$ in candidate set $\mathcal{P}$, we keep $\mathcal{G}$ whose image number $I_v$ or text length $I_t$ satisfy $I_v >= Q'_v$ or $I_t >= Q'_t$, and remove all samples $(x_i, y_i)$ in that group from $\mathcal{D}$. Otherwise, we remove non-satisfied $\mathcal{G}_i$ from $\mathcal{P}$. Ultimately, $\mathcal{P}$ becomes our target set, and $\mathcal{D}$ becomes our updated dataset for the next iteration.

The sampling and filtering stages alternately are repeated for a maximum of $T$ times. The candidate set is acquired $\mathcal{P}$ each time, which includes more valid sample groups $\mathcal{G}$. Meanwhile, we have the updated dataset $\mathcal{D}$ consisting of unselected samples, which is used for sampling and filtering in the next iteration. To ensure that the mini-batches constructed by the ISF method achieve lower Pad Ratio and Dist ratio, appropriate values for $Q_v$ and text $Q_t$ need to be determined. The optimal values for $Q_v$ and $Q_t$ vary across different datasets. In practice, A statistical approach described in Section 5.1 is used to determine these values.

## 4.2 BALANCED MODEL PARTITIONING

Given the number of layers $L$ in the model and the pipeline parallel size $N$, our goal is to find an optimal partition strategy $P = (P^{(1)}, P^{(2)}, P^{(3)}, \ldots, P^{(N-1)})$ such that the training speed of the model is maximized. Here, $P^{(1)} < P^{(2)} < P^{(3)} < \ldots < P^{(N-1)}$, and the $i^{th}$ partition stage $S_i$ consists of layers $l_k$, where $P^{(i-1)} \leq l_k < P^{(i)}$, with $P^{(0)} = 1$ and $P^{(N)} = l + 1$. For example,

| **Algorithm 1** ISF: Sampling Stage | **Algorithm 2** ISF: Filtering Stage |
|---|---|
| 1: $\mathcal{D}$ = randperm($\mathcal{D}$), set $\mathcal{G} = [\ ]$ | 1: Get $\mathcal{P}$ from Sampling Stage |
| 2: **for** $(x_i, y_i)$ in $\mathcal{D}$ **do** | 2: **for** $\mathcal{G}$ in $\mathcal{P}$ **do** |
| 3: $\quad \mathcal{G} \leftarrow \mathcal{G} + (x_i, y_i)$ | 3: $\quad$ **if** $I_v < Q'_v$ and $I_t < Q'_t$ **then** |
| 4: $\quad$ **if** $I_v > Q_v$ or $I_t > Q_t$ **then** | 4: $\quad\quad \mathcal{P} \leftarrow \mathcal{P} - \mathcal{G}$ |
| 5: $\quad\quad \mathcal{G} \leftarrow \mathcal{G} - (x_i, y_i)$ | 5: $\quad$ **else** |
| 6: $\quad\quad \mathcal{P} \leftarrow \mathcal{P} + \mathcal{G}$ , set $\mathcal{G} = [(x_i, y_i),]$ | 6: $\quad\quad$ remove all $(x_i, y_i)$ of $\mathcal{G}$ from $\mathcal{D}$ |
| 7: $\quad$ **end if** | 7: $\quad$ **end if** |
| 8: **end for** | 8: **end for** |
| 9: **return** $\mathcal{P}$ | 9: **return** $\mathcal{P}, \mathcal{D}$ |

given a model with $L = 20$ layers and pipeline size $N = 4$, assume that we have an optimal partition $P = (5, 10, 15)$. The first partition $S_i$ consists of layers $l_1, l_2, l_3, l_4$ since $P^{(0)} = 1, P^{(1)} = 5$.

However, achieving balanced pipeline partitioning for VLMs is a more challenging task compared to LLMs. We must consider: *(1) Model Heterogeneity:* The structural differences between visual and language models make simple parameter-based or layer-based partition strategies ineffective. *(2) Communication Overheads:* Different partitioning strategies result in varying communication volumes, as the number of activations in each layer can differ significantly in VLMs. *(3) Hardware Variability:* Different platforms exhibit varying levels of capability, particularly in terms of communication overhead. On platforms with high network bandwidth, communication overhead can be negligible. Based on the above analysis, A heuristic search algorithm to find the optimal partition is developed. We first identify a candidate set of partition strategies $\{P_k = (P_k^{(1)}, P_k^{(2)}, P_k^{(3)}, \ldots, P_k^{(N-1)}) \mid k = 1, 2, 3, \ldots\}$ that possibly contain the optimal one. Then, the optimal partition strategy $P^*$ is selected by evaluating the actual running time:

$$P^* = \underset{P_i}{\arg \min}\, f(P_i) \tag{3}$$

Here, $f(P_i)$ is the average running time obtained by training the model for several iterations.

**Partition Candidates:** We start by profiling each layer's computation time $\text{FWD}(l_i)$. A greedy algorithm is employed to compute the anchor partition strategy $P^+$, making the computation time of all partition stages $S_i$ close. Around $P^+$, A candidate set of partition strategies is created by jittering $P^{(1)}, P^{(2)}, \ldots, P^{(N-1)}$ within a radius of $r$. When $r = 1$ and $N = 4$, there are a total of $3^3 = 27$ candidates.

**Partition Metrics:** When $r$ and $N$ are very large, there will be a vast number of partition candidates, making it inefficient to evaluate the running time for each one. Therefore, two metrics to rank these candidates are designed.

The first metric is the difference in running time between different pipeline stages $S_i$. Smaller differences generally result in fewer bubbles and faster execution. We use the variance of the running times of different pipeline stages to measure this difference.

$$\text{VAR(fwd\_time)} = \sum_{i=1}^{N} (\text{FWD}(S_i) - \overline{\text{FWD}(S_i)})^2 \tag{4}$$

The second metric is the total point-to-point communication volume of the partition strategy $P_i$. It depends on $P_i$ consisting of $(P^{(1)}, P^{(2)}, P^{(3)}, \ldots)$

$$\text{SUM(comm)} = \sum_{i=1}^{N-1} \text{ACTIV}(l_{pi}) \tag{5}$$

Where $l_{pi}$ is the last layer of partition strategy $P^{(i)}$ and $\text{ACTIV}(l_{pi})$ is the activation number of layer $l_{pi}$, indicating the point-to-point communication volume of $P^{(i)}$. We use the sum of VAR(fwd\_time) and SUM(comm) as the metric for the partition and rank them to select the top $K$ candidates for speed evaluation.

### 4.3 BALANCED ADAPTIVE RE-COMPUTATION

Thanks to the balanced dynamic mini-batch and balanced model partition, a balanced computational load is maintained across each pipeline stage. The memory requirements are now stabilized as the computational demand has been fixed. As a result, we can optimize the re-computation strategy based on actual memory needs, rather than relying on the most aggressive approach to avoid crashes. Reducing the number of re-computations accelerates the model's backward pass, leading to a great improvement in training speed.

We find that heterogeneous architectures have different memory requirements. For example, the vision model in InternVL-Chat-1.5 requires more GPU memory than the language model under the same computational load. Therefore, it is necessary to analyze the memory requirements of each layer in the vision and language models individually and adaptively determine the optimal re-computation strategy for each layer. Specifically, we start by profiling to determine the memory requirements of each layer. Based on the available memory of each device, we then determine the optimal re-computation configurations in pipeline stage $S_i$. More details are shown in A.1.1.

## 5 EXPERIMENTS

In this section, The models and datasets are introduced. Then, we demonstrate the acceleration compared to current state-of-the-art VLMs. Subsequently, a detailed comparison of each component proposed in our method is presented, highlighting its specific contribution to training acceleration. Finally, extensive experimental analysis is conducted.

### 5.1 EXPERIMENTAL SETUP

**Model & Dataset setting:** We conduct experiments following the open-source InternVL-Chat-1.5 setting. Our vision and language models are InternViT-6B and InternLM2-20B, respectively. Two configurations are employed: InternVL-Chat-1.5 (6+20B) and InternVL-Chat-1.5-Plus (6+34B). As the InternVL-Chat-1.5 dataset is not yet available, we utilize the InternVL-Chat-1.2 dataset, which comprises approximately 1.2 million samples, as an alternative. All other training settings remain unchanged. GPU Days are used as our evaluation metric to estimate the total training time. Specifically, GPU Days are reported based on A100 GPU usage to evaluate the speed-up performance.

**Implementation Details:** We determine $Q_v$ and $Q_t$ by using statistics of datasets. First, we traverse the entire dataset and collect the summation of the lengths of all text tokens and the number of images. Then, We calculate the average number of text tokens per image. We set $Q_t$ as the length of the longest text token in the dataset and use the calculated text-to-image ratio to determine $Q_v$. For images, we set $Q'_v = Q_v$, and for text, we set $Q'_t = Q_t - 128$. In the InternVL-Chat-1.2 dataset, $Q_t = 4K$, $Q_v = 9$. Note that $Q_v$ refers to the number of images. Each image will be processed into 1K tokens before being fed into VIT.

### 5.2 MAIN RESULTS

We demonstrate the superiority of our method under various settings in Table 2. Our baseline model is InternVL-Chat-1.5 (6+20B) Chen et al. (2024), utilizing DeepSpeed ZeRO-3 as the training backend. OmniBal reduces GPU days from 38.9 to 25.3, achieving a 1.54x speed-up. Simultaneously, we consistently maintain comparable performance across commonly used datasets, such as MMB-EN/CN Liu et al. (2023c), ChartQA Masry et al. (2022), AI2D Kembhavi et al. (2016), MMVet Yu et al. (2023), and MME Fu et al. (2023).

Experiments with Megatron-DeepSpeed are conducted, which integrates tensor, pipeline, and data parallelism for larger-scale models. However, directly applying 3D parallelism can slow down training due to the heterogeneous nature of VLM models. Table 2 shows that switching to Megatron-DeepSpeed increased GPU days from 38.9 to 61.8. OmniBal addresses this issue by achieving computational balance across data, model, and memory, reducing GPU days from 61.8 to 21.3. This demonstrates the importance of computational balance for effective 3D parallelism. Notably, our method also outperformed DeepSpeed, highlighting the superiority of 3D parallelism when balanced computation is achieved. Results under a larger-scale setting (InternVL-Chat-1.5-Plus) are

Table 2: Main Results. We use open-source InternVL-Chat-1.5 6+20B and 6+34B as the models with either DeepSpeed (ZeRO-3) or Megatron-Deepspeed backend. GPU Days are reported in the InvernVL-Chat-1.2 1.2M training dataset to show the speed-up ratio. Models are also evaluated on five commonly used benchmarks.

| Model | Balance? | Backend | MMB-EN/CN | ChartQA | AI2D | MMVet | MME | GPU Days (speed-up) |
|-------|----------|---------|-----------|---------|------|-------|-----|---------------------|
| 6+20B | × | DeepSpeed | 78.2/77.4 | 86.2 | 71.3 | 48.9 | 1901.2 | 38.9 (1x) |
|       | ✓ | DeepSpeed | 78.7/77.6 | 86.5 | 71.4 | 50 | 1969.4 | 25.3 (**1.54x**) |
|       | × | Megatron | 79.5/77.7 | 87.3 | 71.6 | 45.0 | 1957.7 | 61.8 (0.63x) |
|       | ✓ | Megatron | 78.6/77.5 | 86.7 | 70.9 | 48.5 | 1956.3 | 21.3 (**1.83x**) |
| 6+34B | × | DeepSpeed | 80.0/79.2 | 86.6 | 73.4 | 45.9 | 2015.8 | 54.3 (1x) |
|       | ✓ | DeepSpeed | 80.9/79.0 | 89.1 | 73.3 | 47.0 | 2153.6 | 35.5 (**1.53x**) |
|       | × | Megatron | 80.2/79.3 | 88.9 | 73.7 | 44.2 | 2111.9 | 75.4 (0.72x) |
|       | ✓ | Megatron | 80.1/78.0 | 89.3 | 73.5 | 45.4 | 2072.7 | 30.5 (**1.8x**) |

Table 3: Ablation studies of components

| data balance | model balance | memory balance | GPU Days |
|--------------|---------------|----------------|----------|
|              |               |                | 61.8 |
| ✓            |               |                | 51.9 |
| ✓            | ✓             |                | 29.0 |
| ✓            | ✓             | ✓              | **21.3** |

Table 4: Results on different datasets

| Dataset | Dist Ratio | | GPU Days |
|---------|------------|------|----------|
|         | *VIT* | *LLM* | |
| LLava-665K | 0.02 | 0.145 | 43.3→ 12.4 |
| InternVL-1.2M | 0.02 | 0.14 | 61.8→ 21.3 |
| LCS-558K | 0.001 | 0.029 | 23.8→ 7.5 |

also reported to verify the generalizability of our method. The larger model consistently improves, accelerating the training process while maintaining model performance.

## 5.3 ABLATION ANALYSIS

In this section, ablation experiments on each component of our method are conducted, using InternVL-Chat-1.5 as the baseline model with a 3D parallel Megatron-DeepSpeed backend. Table 3 illustrates the impact of each component. The baseline model experiences a considerable slowdown in training speed due to computational imbalance, necessitating a total of 61.8 GPU days. By achieving data balance, GPU days are reduced to 51.9. Data balance allows us to achieve a more balanced model partition, reducing the training time. Finally, optimizing memory with an adaptive re-computation strategy reduces GPU days to 21.3. These results demonstrate that a holistic balance encompassing data, model, and memory is crucial for efficient VLM training. Below we provide a detailed analysis of each component.

**The Importance of Data Balance:** In Table 5, we investigate the importance of data balance in large-scale distributed training by comparing four methods: (1) Baseline: Randomly combining data into a mini-batch with padding aligned to the longest input within mini-batches (2) Length-Group: Combining samples with similar text and image sizes into a mini-batch to minimize padding within mini-batch. (3) Device-Group: Grouping samples with similar input sizes across devices to minimize idle times. (4) Balanced Dynamic Mini-batch: Using ISF to construct balanced mini-batches within mini-batches and cross devices.

Table 5 reveals the following: (1) Baseline: is the slowest due to the completely random combination of different-sized samples, leading to significant size variation and excessive padding (0.31). Meanwhile, high Dist Ratio ViT (0.34) and LLM (0.30) result in computation disparities between devices, severely impacting throughput efficiency. (2) Length-Group: enhances throughput efficiency by pre-grouping samples of similar sizes into mini-batches, thus reducing the internal padding ratio (0.2). Minimizing the number of redundant tokens within mini-batches effectively lowers the GPU days required to 54.0. (3) Device-Group: reduce idle time by ensuring consistent input sizes across devices. It decreases the Dist Ratio of ViT (0.125) and LLM (0.15). However, it only balances input sizes between devices and neglects the balance within mini-batches. High padding (0.378) wastes computational resources. (4) Our Approach: balances input sizes within mini-batches on each de-

Table 5: Importance of data balance. AVE-BS indicates the average batch size in each iteration. We report results with Model Balance (MB) and without MB.

| Method | AVE-BS | Max-Seq-Len | | Pad Ratio | Dist Ratio | | Balanced | GPU Days | |
| --- | --- | --- | --- | --- | --- | --- | --- | --- | --- |
| | | *VIT* | *LLM* | | *VIT* | *LLM* | | *w/o* MB | *w* MB |
| baseline | 4 | 20K | 16K | 0.31 | 0.34 | 0.30 | × | 61.8 | 42.2 |
| length-group | 4 | 20K | 16K | 0.20 | 0.26 | 0.13 | × | 54.0 | 40.0 |
| device-group | 4 | 20K | 16K | 0.378 | 0.125 | 0.15 | × | 54.5 | 43.6 |
| **ISF(ours)** | 4.6 | **9K** | **4K** | **0** | **0.02** | **0.14** | ✓ | **51.9** | **29.0** |

Table 6: Importance of model balance. VAR indicates variance. SUM(comm) is the summation of commutation volume (MByte)

| Method | VAR(param) | VAR(num_layer) | VAR(fwd_time) | Δ SUM(comm) | GPU days |
| --- | --- | --- | --- | --- | --- |
| (1) parameter-based | **0.03** | 13.4 | 93.6 | +0.0 | 42.2 |
| (2) layer-based | 0.64 | **1.2** | 20.1 | +8.2 | 30.6 |
| (3) profile-based | 0.85 | 2.1 | **6.5** | +16.6 | 30.9 |
| (4) **BMP (ours)** | 0.83 | 1.5 | 12.2 | -21.0 | **29.0** |

vice and across devices simultaneously. It reduces both the Pad Ratio and the Dist Ratio, achieving a padding ratio of 0 while maintaining a lower Dist Ratio of 0.02 and 0.14. While our method balances input sizes, model partitioning still limits training speed. With model balance (MB), GPU days are reduced from 42.2 to 29.0, a gain of 13.2, compared to 9.9 without MB (from 61.8 to 51.9). This underscores the importance of a holistic balance approach.

**The Importance of Model Balance:** Table 6 examines balanced model partitioning, focusing on partition strategies for pipeline parallelism. For LLM training, common methods include (1) parameter-based and (2) layer-based, (3) profile-based methods such as DreamPipe Narayanan et al. (2019) estimate the computation time for each layer and use this information to partition the model effectively. Additionally, (4) our search-based Balanced Model Partition method finds the optimal partition strategy from a set of candidates. As shown in Table 6, (1) Parameter-based and (2) layer-based methods split the model's parameters or layers across devices, achieving low variation in VAR(param) and VAR(num_layer). However, they still show high variation in forward time VAR(fwd_time), leading to computational inefficiencies in the pipeline. (3) The profile-based method ensures the optimal VAR(fwd_time). However, this partitioning occurs before the vision model's token sub-sampling operation, increasing communication overhead and affecting training speed. (4) Our proposed Balanced Model Partition (BMP) method explores a high-quality partition strategy space to identify the optimal strategy, achieving the best results in 29.0 GPU days.

**The Importance of Memory Balance:** In Table 7, we examine the significance of memory balance. In the baseline model, varying input sizes for vision (4K–20K tokens) and language (1K–16K tokens) lead to varying GPU memory usage. Despite aggressive re-computation, the remaining memory on an 80G A100 can drop to 7.3G. ISF and BMP improve training speed by controlling computational load across devices. However, memory demands still varied, *e.g.*, GPUs 1 and 2 having more remaining memory. Our method further improves training speed by adjusting the re-computation strategy to fully utilize the remaining memory, reducing GPU days to 21.3.

## 5.4 COMPONENT ANALYSIS

**Convergence of ISF:** The convergence performance of ISF is evaluated, with the results illustrated in Figure 3. On the LLava-665K dataset Liu et al. (2023a), we observe that the Dist Ratio for both vision and language data dropped significantly after just one iteration. After five iterations, the Dist Ratio stabilized considerably. In practice, we perform ten iterations to ensure stable results, which only take less than one minute. The computational cost is negligible relative to the overall runtime. Additionally, our method is tested on two other datasets, InternVl-1.2M Chen et al. (2024) and LCS558K Liu et al. (2023b), and observed similar convergence rates.

Table 7: Importance of memory balance. $VRAM_i$ denotes remaining VRAM(G) in pipeline stage $S_i$. For the baseline model, the metric varies as $<$minimum$> \sim <$maximum$>$.

| Method | V-Seq-Len | L-Seq-Len | $VRAM_1$ | $VRAM_2$ | $VRAM_3$ | $VRAM_4$ | GPU Days |
|---|---|---|---|---|---|---|---|
| baseline | 4K~20K | 1K~16K | 13~50.2 | 7.3~40.5 | 7.3~40.5 | 7.3~40.5 | 61.8 |
| + data & model balance | 9K | 4K | 58.2 | 56.2 | 32.5 | 32.7 | 29.0 |
| + memory balance | 9K | 4K | 12.3 | 21.7 | 24.7 | 30.0 | **21.3** |

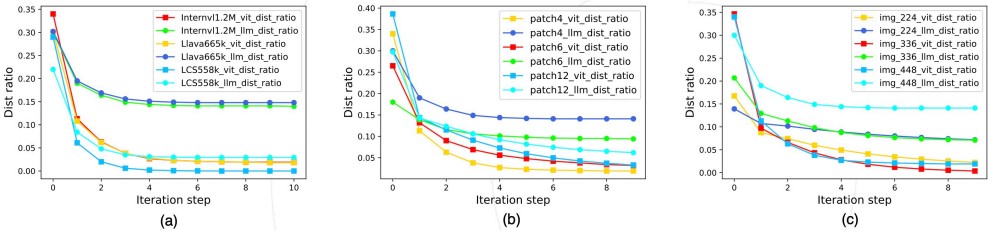

Figure 3: ISF convergence testing. We test the convergence of ISF in various scenarios, including (a) different datasets, (b) different patch sizes, and (c) different image resolutions.

**Generalization Capability**: We study the generalization capability of our method from multiple aspects: (1) Different Datasets: As shown in Table 4, we achieved consistently low Dist Ratio on LLava-665K, InternVL-1.2M, and LCS558K and significantly improved training speed. (2) Different Models: Experiments are conducted using various combinations of vision and language models. For vision models, in addition to InternVL-6B, the open-source EVA-CLIP models is incorporated, which span a range from 1B parameters Sun et al. (2023a) to 18B parameters Sun et al. (2023b). On the language side, several models are utilized, including Llama3-8B, Llama3-70B AI (2024), Yi-34B NousResearch (2023), and the large-scale Qwen1.5-110B Bai et al. (2023a). As detailed in Appendix A.2, our approach significantly reduces the GPU days required for model training. (3) Different High-Resolution Setting: Under various settings, we achieved a speedup of approximately 2.0x, as demonstrated in Appendix A.3. (4) Different Tasks: Besides SFT tasks, pretraining tasks are also tested, as shown in Appendix A.4, and we observed consistent improvements across all settings. (5) Different Image Resolutions: As shown in Appendix A.5, our method consistently delivers a highly satisfactory acceleration effect with different input image resolutions. (6) Different Model-series: As demonstrated in Appendix A.6, our approach also achieves significant acceleration with LLava-1.6. (7) Pre-Processing Strategy: Qwen2-VL team (2024) employs a novel pre-processing strategy to support native dynamic resolution. We utilized this approach in ablation studies and achieved comparable acceleration effects (approximately 1.9x in Appendix A.7). (8) Long-Context Support: The capability to handle long-context is crucial for multi-modal foundation models. Our balanced solution is also applicable to long-context training using sequence parallelism. Further details can be found in Appendix A.8. (9) Different Hardware Results: Appendix A.9 presents the efficiency of our method across various hardware platforms, including different GPUs (e.g., A100, H100) and network bandwidths. (10) Large-Scale Results: large-scale experiments are shown in Appendix A.10 on 512 GPUs and our method is still effective. These results underscore the effectiveness and robustness of our method across a wide range of datasets, models, and tasks.

# 6 CONCLUSION

In this work, we effectively addressed the issue of imbalanced computation loads in large-scale 3D parallel training of vision-language models by rebalancing across data, model, and memory dimensions. Experimental results demonstrate that our method can significantly speed up training on many open-source models. The effectiveness and generalizability of our approach are also validated across various models, datasets, and hardware platforms. Our method can accelerate the development of this field by enabling more efficient training.

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
