# A  APPENDIX

## A.1  TRAINING-PIPELINE OF OUR BALANCED METHOD

### A.1.1  BALANCED ADAPTIVE RE-COMPUTATION STRATEGY

Here, we provide a detailed introduction to the Balanced Adaptive Re-Computation Strategy. In this context, $Q_v$ and $Q_t$ represent the inputs for Vision Transformer (VIT) and Large Language Model (LLM) respectively. $M_r$ denotes the remaining GPU memory at the current stage, while $M_t$ and $M_v$ indicate the GPU memory saved by each transformer layer of the LLM and VIT when enabling re-computation.

**Step-1**: Given the inputs $Q_v$ and $Q_t$, we enable the re-computation strategy across all transformer modules of the model. At each forward pass, we clear the cache and record each stage's remaining memory usage $M_r$.

**Step-2**: We manually disable re-computation for some layers based on the remaining GPU memory. Subsequently, we record the GPU memory usage $M_r'$ for each stage.

**Step-3**: Based on the memory differences $\Delta M_r$ observed between Step-1 and Step-2, along with the re-computation strategy implemented at each stage, we estimate the memory savings $M_t$ and $M_v$ for each transformer layer of the VIT and LLM, respectively.

**Step-4**: Based on the estimated GPU memory savings $M_t$ and $M_v$ measured in Step-3, as well as the remaining memory $M_r$ from Step-1, We first estimate the theoretically optimal re-computation strategy for each stage and conduct the training test. If the test runs successfully, we adopt this strategy. If it fails, we incrementally increase the number of re-computation layers by the remaining GPU memory for each stage.

### A.1.2  TRAINING-PIPELINE

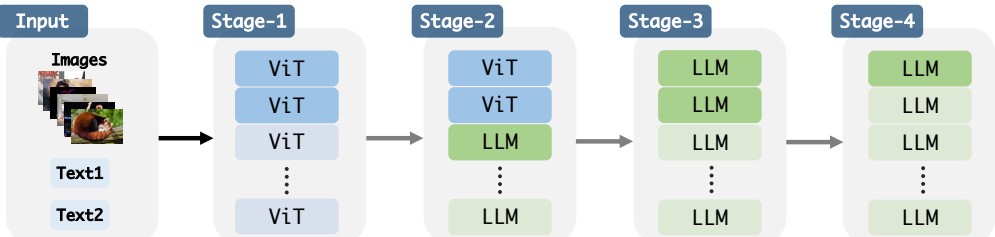

Figure 1: The pipeline consists of four stages, labeled Stage-1 to Stage-4, each representing a different stage of pipeline parallelism. Within this structure, "ViT" stands for the Vision Transformer layer, while "LLM" refers to the Transformer layer used for large language models (LLM). Regarding computational execution, the darker-colored sections signify forward passes with re-computation. In contrast, the lighter-colored sections denote a standard forward pass without re-computation.

## A.2  RESULTS ON DIFFERENT MODEL SIZE

We test various combinations of vision and language models. As shown in Table 1, our approach significantly reduces the required GPU days for model training, achieving nearly a 2x speedup across models of various sizes.

## A.3  RESULTS ON DIFFERENT DYNAMIC HIGH-RESOLUTION SETTING

To validate the effectiveness of our method, we test it under various high-resolution settings. Our approach consistently demonstrates low Dist Ratio and strong acceleration across all configurations as shown in Table 2, significantly improving training speed under different settings.

Table 1: Results for different model sizes are shown, with TP, PP, and DP representing various distributed training strategies: Tensor Parallel (TP), Pipeline Parallel (PP), and Data Parallel (DP), respectively. The "Stages-Layer-Num (V+L)" column indicates the number of Vision Transformer (V) and Language Transformer (L) layers assigned to each stage. Additionally, the "Re-computation" column denotes the number of re-computations enabled in each stage.

| Vision-Model | Language-Model | TP PP DP | Stages-Layer-Num (V+L) | Re-computation | GPU Days(speed up) |
|---|---|---|---|---|---|
| InternVL-6B | Llama3-8B | (1,4,8) | [16,17,20,24] | [8,7,10,24] | 27.7 → **13.8(2.0x)** |
| InternVL-6B | InternLM2-20B | (2,4,4) | [22,23,24,24] | [0,0,0,0] | 61.8 → **21.3(2.9x)** |
| InternVL-6B | Yi-34B | (4,4,2) | [28,29,24,24] | [3,2,0,0] | 75.4 → **30.5(2.5x)** |
| InternVL-6B | Llama3-70B | (4,8,2) | [22,23,13,14,14,14,13,12] | [11,12,8,5,3,2,0,0] | 129 → **52.5(2.4x)** |
| InternVL-6B | Qwen1.5-110B | (8,8,1) | [21,22,13,13,14,14,14,14] | [6,9,1,3,0,0,0,0] | 243 → **75.2(3.2x)** |
| EVA-CLIP-1B | InternLM2-20B | (2,4,4) | [43,16,15,14] | [0,0,0,0] | 23.6 → **12.2(1.9x)** |
| EVA-CLIP-4B | InternLM2-20B | (2,4,4) | [39,22,21,20] | [10,8,1,3] | 38.1 → **17.0(2.2x)** |
| EVA-CLIP-8B | InternLM2-20B | (2,4,4) | [17,18,23,22] | [5,5,8,10] | 41.8 → **20.3(2.0x)** |
| EVA-CLIP-18B | InternLM2-20B | (4,4,4) | [18,19,25,34] | [2,2,0,0] | 63.6 → **33.8(1.9x)** |

Table 2: Results on different dynamic high-resolution settings. "Max-Patch-Num" indicates the maximum number of patches into which an image can be divided. This parameter controls the granularity of image segmentation, impacting both model performance and computational efficiency. Adjusting the Max-Patch-Num allows for more flexible handling of high-resolution images in the model, optimizing resource usage while maintaining accuracy.

| Model | Max-Patch-Num | AVE-BS | Max-Seq-Len | | Dist Ratio | | GPU Days (speed-up) |
|---|---|---|---|---|---|---|---|
| | | | *VIT* | *LLM* | *VIT* | *LLM* | |
| InternVL-6B-20B | 1 | 7.6 | 9K | 4K | 0.06 | 0.05 | 28.6 → 13.7 (2.1x) |
| | 4 | 4.6 | 9K | 4K | 0.02 | 0.14 | 61.8 → 21.3 (2.9x) |
| | 6 | 2.7 | 14K | 5K | 0.019 | 0.136 | 147 → 72 (2.05x) |
| | 12 | 1.9 | 14K | 5K | 0.03 | 0.12 | 209 → 105 (2.0x) |

## A.4 RESULTS ON PRETRAIN SETTING

We evaluate our method in other tasks like Pre-training task. In Pre-training, we train both the Vision Transformer (ViT) and MLP components for models ranging from 6B to 20B. However, for larger models, such as 6B-34B and 6B-70B, we focus solely on training the MLP component. Across all configurations, we observe consistent performance improvements shown in Table 3, particularly with the largest model, where GPU days are significantly reduced from 16.8 to 9.6, demonstrating enhanced training efficiency.

Table 3: Results on Pretrain Setting

| Model | Dataset | Trainable Module | AVE-BS | Dist Ratio | | GPU Days |
|---|---|---|---|---|---|---|
| | | | | *VIT* | *LLM* | |
| InternVL-6B-20B | LCS-558K | ViT+MLP | 5.8 | 0 | 0.03 | 9.9 → 6.0 (1.65x) |
| InternVL-6B-34B | LCS-558K | MLP | 5.1 | 0 | 0.031 | 8.3 → 4.9 (1.69x) |
| InternVL-6B-70B | LCS-558K | MLP | 5.2 | 0 | 0.029 | 16.8 → 9.6 (1.75) |

## A.5 RESULTS ON DIFFERENT RESOLUTIONS

We further test our method with different image resolution inputs. As shown in Table 4, our method consistently delivers low Dist Ratio and highly satisfactory acceleration results across varying image resolutions, demonstrating its effectiveness in improving training efficiency.

**Table 4: Results on Different Resolutions**

| Resolution | AVE-BS | Dist Ratio | | GPU Days |
|------------|--------|------------|------|----------|
| | | *VIT* | *LLM* | |
| 224 | 4.8 | 0.009 | 0.068 | 32.0 → 20 (1.6x) |
| 336 | 3.3 | 0.005 | 0.07 | 62.0 → 33 (1.88x) |
| 448 | 4.6 | 0.02 | 0.14 | 61.8 → 21.3 (2.9x) |

## A.6 RESULTS ON OPEN-SOURCE LLAVA-1.6

We also validate our method using another popular open-source model, LLava-1.6, with the Deep-Speed backend, as shown in Table 5. For the DeepSpeed backend, we employ only our balanced dynamic mini-batch strategy. In the case of the open-source LLava model, while its ViT component is relatively small and the imbalance occurs primarily at the data level, we still achieved a notable overall speedup. Although the speedup ratio is smaller compared to other models, our method delivered a 30% improvement in performance.

**Table 5: Results on Open-source LLava-1.6**

| Model | AVE-BS | Dist Ratio | | GPU Days |
|-------|--------|------------|------|----------|
| | | *VIT* | *LLM* | |
| Llava-1.6-7B | 4.54 | 0.008 | 0.037 | 10.2 → 7.7 (1.3x) |
| Llava-1.6-13B | 4.54 | 0.008 | 0.037 | 18 → 13.3 (1.35x) |
| Llava-1.6-34B | 4.4 | 0.009 | 0.0041 | 42.7 → 31.3 (1.36x) |

## A.7 RESULTS ON QWEN2-VL PRE-PROCESSING STRATEGY

Qwen2-VL is a recent, highly regarded open-source project that provides strong support for dynamic image input. Consequently, we adopt the pre-processing strategy of Qwen2-VL to validate our method. As shown in Table 6, our approach demonstrates a substantial acceleration effect (approximately 1.9x) when applied to the Qwen2-VL strategy, significantly reducing both the padding ratio and dist ratio.

**Table 6: Results on Qwen2-VL Pre-Processing strategy**

| Model | Dataset | AVE-BS | Pad-Ratio | Dist Ratio | | GPU Days (speed-up) |
|-------|---------|--------|-----------|------------|------|---------------------|
| | | | | *VIT* | *LLM* | |
| InternVL-6B-20B | InternVL-1.2M | 4 | 0.31 | 0.408 | 0.393 | 40.2 (1x) |
| InternVL-6B-20B | InternVL-1.2M | 6.6 | 0 | 0.12 | 0.06 | 21.0 (1.9x) |

## A.8 LONG-CONTEXT RESULTS

Our method can also be applied to long-context training. To evaluate its effectiveness, we constructed a dataset named Long-2.5W consisting of multi-modal inputs with a maximum text length of 32k tokens and up to 80 images. Handling both long and short texts together is often necessary in long-context training. Thus, it's essential to maintain balance not only at the data-parallel level but also at the sequence-parallel level.

To address this, we propose a straightforward solution. For long text inputs, we evenly split the images across different sequence-parallel (SP) processes, and then gather them during LLM training. For short multi-modal training samples, we apply our balanced group ISF algorithm, which ensures that both sequence and data parallelism remain approximately balanced. Additionally, we designed a grouping sampler to ensure that long and short multi-modal text samples remain relatively independent at the data-parallel level. Figure 2 illustrates our complete training pipeline.

In this instance, we set sequence parallelism to 4. To maintain the original InternVL-1.2M input at 32k, we expand the training input batch size to 10. As shown in Table 7, our method achieves nearly identical training speeds for both separate and mixed training. Compared to basic hybrid training, our approach substantially reduces the dist ratio while significantly accelerating the training process.

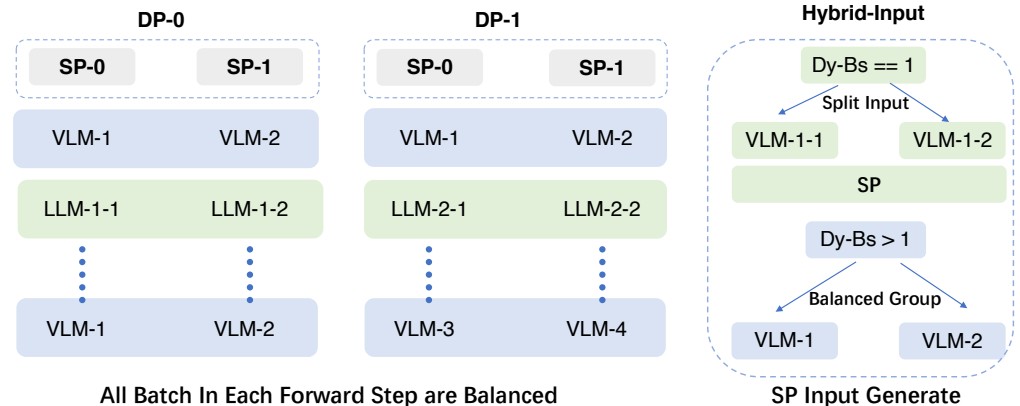

Figure 2: SP-0 and SP-1 denote different sequence parallel process numbers, while DP-0 and DP-1 represent different data parallel process numbers. VLM-1-1 and VLM-1-2 refer to the two resulting inputs after splitting the same input, whereas VLM-1 and VLM-2 correspond to two distinct sets of inputs.

Table 7: Results on Long-Context Training

| Dataset | AVE-BS | Max-Seq-Len | | Pad-Ratio | Dist Ratio | | SP-Ratio | GPU Days |
|---|---|---|---|---|---|---|---|---|
| | | VIT | LLM | | VIT | LLM | | |
| InternVL-1.2M | 8 | 40k | 32k | 0.417 | 0.27 | 0.24 | 0 | 36.2 |
| InternVL-1.2M-Balanced | 10.3 | 20K | 8K | 0 | 0.03 | 0.07 | 0 | 18.7 |
| Long-2.5W | 1 | 80K | 32K | 0 | 0.03 | 0.02 | 0.22 | 23.2 |
| Long-2.5W-Balanced | 1 | 80K | 32K | 0 | 0.03 | 0.02 | 0 | 19.6 |
| InternVL-1.2M + Long-5W | 3.9 | 80K | 32K | 0 | 0.03 | 0.08 | 0.025 | 38.6 |

## A.9 DIFFERENT HARDWARE RESULTS

We test our method on various hardware platforms with different GPUs (e.g., A100, H100) and network bandwidths. The experiments in Table 8 confirmed consistent performance improvements across all platforms.

Table 8: Results on Different Hardware. IB indicates network bandwidths

| Dataset | Hardware | IB | Dist Ratio | | GPU Days (speed-up) |
|---|---|---|---|---|---|
| | | | VIT | LLM | |
| InternVL-1.2M | A100 | 4x200G | 0.02 | 0.145 | 61.8 → 21.3 (2.90x) |
| InternVL-1.2M | A100 | 2x200G | 0.02 | 0.145 | 64.0 → 24.8 (2.58x) |
| InternVL-1.2M | H100 | 8x400G | 0.02 | 0.145 | 32.5 → 12.2 (2.67x) |

## A.10 LARGE-SCALE RESULTS

To validate the effectiveness of our method, we conduct a study using larger-scale models and a greater number of GPUs. As shown in the Tabel 9, our method achieves a speedup ratio exceeding 2.0 across varying GPU configurations. Moreover, the results demonstrate that our approach maintains a more favorable linear speedup (85% → 95%) as GPUs increase.

Table 9: Results on Large-Scale models (6 + 70B) and GPUs

| Dataset | Hardware | IB | GPUs | Dist Ratio | | GPU Days (speed-up) |
|---------|----------|-----|------|------|------|---------------------|
| | | | | *VIT* | *LLM* | |
| InternVL-1.2M | H100 | 8x400G | 64 | 0.02 | 0.139 | 72.8 → 29.3 (2.48x) |
| InternVL-1.2M | H100 | 8x400G | 128 | 0.02 | 0.139 | 75.2 → 29.7 (2.53x) |
| InternVL-1.2M | H100 | 8x400G | 256 | 0.02 | 0.139 | 82.1 → 30.4 (2.70x) |
| InternVL-1.2M | H100 | 8x400G | 512 | 0.02 | 0.139 | 85.3 → 30.9 (2.76x) |