# OpenReview forum: "OMNIBAL: TOWARDS FAST INSTRUCT-TUNING FOR VISION-LANGUAGE MODELS VIA OMNIVERSE COMPUTATION BALANCE"
_ICLR.cc/2025/Conference — ICLR 2025 Conference Withdrawn Submission_

### Official Review · Reviewer_Fc4X · 2024-10-30

**Soundness:** 3
**Presentation:** 3
**Contribution:** 3
**Rating:** 6
**Confidence:** 3

**Summary:**

This paper aims at addressing the issue of imbalanced computation loads in large-scale 3D parallel training of vision-language models. It rebalances across data, model, and memory dimensions. Experimental results demonstrate that this method can effectively speed up training on many open-source models.

**Strengths:**

Strength：

1. Extensive experiments demonstrate the approach's effectiveness for rebalancing computation for MLLMs.
2. The motivation is clear and the in-depth analysis and ablations are convincing.
3. This paper is well-organized and clearly-written.

**Weaknesses:**

N/A

**Questions:**

N/A

---

> ### Author Response · Authors · 2024-11-21
> **Response to Reviewer Fc4X**
>
> Thank you for finding the topic of our work is of increasing importance, and appreciating our contribution.

---

### Official Review · Reviewer_r51z · 2024-11-04

**Soundness:** 3
**Presentation:** 3
**Contribution:** 3
**Rating:** 6
**Confidence:** 3

**Summary:**

This paper focuses on the 3D parallel training of MLLMs. The approach to balancing the computational load concerning data, model, and memory distribution both in-device and cross-device is straightforward yet effective. The proposed method demonstrates significant time reduction while maintaining the model's capabilities.

**Strengths:**

+ The paper centers on the efficient parallel training of MLLMs particularly focusing on balancing the computational load concerning data, model, and memory distribution both in-device and cross-device.

+ The multi-modal sample balancing strategy stands out for its simplicity, offering a straightforward approach to load management. Despite its simplicity, the method proves to be effective, yielding significant reductions in training GPU hours while preserving the model’s capabilities. TItmakes the approach both accessible for large-scale applications.

+ It is comprehensively validated across different datasets, tasks, resolutions and hardware archs etc

**Weaknesses:**

- Results on other MLLM architectures, especially those commonly used in the community such as the LLaVA series, are expected in a technical report style work.

**Questions:**

See weaknesses.

---

> ### Author Response · Authors · 2024-11-19
> **Response to Reviewer r51z**
>
> **W**:  Thank you for your positive evaluation and thoughtful feedback on our work. Appendix A.6 contains the experimental results of the LLava-1.6 series, demonstrating the effectiveness of our method. In Section 5.4 and the Appendix, we also verified its effectiveness in various other settings.

---

### Official Review · Reviewer_dcak · 2024-11-04

**Soundness:** 2
**Presentation:** 2
**Contribution:** 2
**Rating:** 3
**Confidence:** 4

**Summary:**

This paper introduces OmniBal, a training framework that addresses computational imbalances in large-scale Vision-Language Models (VLMs) during 3D parallel distributed training. OmniBal improves efficiency by balancing data, model, and memory across devices. For data, the framework forms balanced mini-batches that reduce padding and minimize computational idle times. For the model, a search-based method is used to optimally partition vision and language components, addressing their inherent structural differences. For memory, adaptive re-computation maximizes memory utilization, enhancing training speed. Experiments show OmniBal reduces training time significantly.

**Strengths:**

1. This paper addresses a critical issue: accelerating distributed training for large-scale vision-language models (VLMs).

2. The authors present a comprehensive optimization framework that targets VLM distributed training from three perspectives: data, model, and memory balance.

**Weaknesses:**

1. The paper lacks sufficient motivation data to illustrate the imbalance problem quantitatively.

2. The contributions of this paper seems incremental.

3. Some figures have fonts that are too small, making them challenging to read and interpret.

Please see my comments for details.

**Questions:**

This paper propose OmniBal, which improves efficiency by balancing data, model, and memory across devices. It achieves significant improvements over baselines. However, I have following questions/concerns:

1. The data partitioning approach in this paper appears to offer limited novelty. Numerous adaptive data batching methods, such as [1] and [2], have already addressed similar issues, albeit without specific evaluations on VLMs. However, applying these techniques to VLMs does not seem particularly challenging, given that (i) the primary goal—balancing computation both within and across batches—is quite similar, and (ii) the methodology, which involves batching based on profiled or predicted latency, closely resembles the existing works. Could the authors clarify the unique contributions of their data partitioning method in comparison to these established batching approaches?

2.  The novelty of the model partitioning and re-computation strategies also seems limited. The AdaPipe framework, as referenced in the paper, performs layer-wise partitioning and adaptive re-computation based on computation and memory costs. Since the information needed for partitioning VLMs can also be easily profiled, could the authors explain the specific advantages of their approach over extending AdaPipe for VLMs?

3. The motivation for addressing data imbalance would be stronger with quantitative results from real datasets illustrating variation in sample sizes or computational demands. The simplified example in Figure 2 does not adequately demonstrate the scope of the issue.

4. The section on memory imbalance in Section 3 lacks clarity. While it states the need for aggressive re-computation, it would benefit from a discussion on potential opportunities to mitigate memory imbalance.

5. The figure in Appendix A.1.2 does not clearly illustrate how computation is balanced, making it difficult to assess the efficacy of the approach.

6. Section 4 introduces Q'v and Q't to reduce Pad Ratio and Dist Ratio, but the paper lacks a clear formula or explanation of how these values affect the ratios. This makes it challenging to understand how Q values are chosen to achieve the intended balance.

7. In Section 5, Q′v is set equal to Qv, while Q′t is defined as Qt − 128. Could the authors provide an intuition or rationale behind these choices?

8. The partitioning approach begins with an anchor partition strategy. What is the computational overhead for generating this anchor partition, and how close is it to the optimal partition? If it is far from optimal, what steps are taken to improve upon it? Conversely, if it is close, is it necessary to generate multiple new candidate partitions?

minor point(s):
1. The fonts in some figures are too small, particularly in Figure 3, where the axis labels and legends are difficult to read.


[1] Yu, Gyeong-In, Joo Seong Jeong, Geon-Woo Kim, Soojeong Kim, and Byung-Gon Chun. "Orca: A distributed serving system for {Transformer-Based} generative models." In 16th USENIX Symposium on Operating Systems Design and Implementation (OSDI 22), pp. 521-538. 2022.

[2] Choi, Yujeong, Yunseong Kim, and Minsoo Rhu. "Lazy batching: An sla-aware batching system for cloud machine learning inference." In 2021 IEEE International Symposium on High-Performance Computer Architecture (HPCA), pp. 493-506. IEEE, 2021.

---

> ### Author Response · Authors · 2024-11-19
> **Response to Reviewer dcak （1）**
>
> Thanks for pointing it out.  We have updated some materials ([revision version](https://openreview.net/pdf?id=N80ER2he6l) changes are marked in blue font), specifically Section 3, to enhance Figures 2 and 3, which now better illustrate the unique computation imbalance problem in VLM instruct-tuning training.
>   First, we introduce the computation imbalance problem in more detail and discuss the difference between LLM training and VLM instruct-tuning training. We then present our motivation and address several concerns.
>
> ## Computation Imbalance Problem
>    The computation imbalance of VLM instruct-tuning training consists of two dimensions: Inter-Stage and Intra-Stage:
>   - **Inter-Stage** imbalance means the computation imbalance of different pipeline parallel stages (stage 1 to stage 4 in **Figure 2**).
>   - **Intra-Stage** imbalance indicates the computation imbalance of the same stage across time and devices.
>   **They both include three specific levels: data, model, and memory.**
>
>   - **Data**:  As shown in Figure 2, at time T-0, the Vision and LLM inputs for DP-0 (Group 0 of data-parallel) and DP-1 (Group 1 of data-parallel) differ (Img=4, Text=2k vs. Img=9, Text=4k). Different inputs will bring different computational complexities. For example, the total forward time of DP-0 at time T-0 is 970ms (500 + 160 + 150 + 160), and the total forward time of DP-1 at time T-0 is 1730ms (900 + 300 + 260 + 270). DP-0 needs to wait for DP-1 to complete all training before updating parameters, which will cause DP-0's GPU resources to wait for a long time. This is the **Intra-Stage imbalance problem across devices**. For DP-0, the inputs at times T-0 and T-1 on the same device also differ significantly (e.g., Img=4, Text=2k versus Img=20, Text=16k), which is an **Intra-Stage imbalance problem** over time. This results in a substantial variance in forward time and memory usage, creating challenges in determining a globally optimal model partition strategy.
>   - **Model**:  For DP-0, at T-0 and T-1, the standard deviation (148ms and 402 ms) of the forward time across different stages is significant, leading to serious bubbles in pipeline parallelism and slowing down training speed. This is the **Inter-Stage imbalance problem**.  Furthermore, the computation distribution across stages shows a huge difference between T-0 and T-1. For example, the computation distribution at T-0 is (500:160:150:160)=（0.52:0.16:0.15:0.16)，while the computation distribution at T-1 is (2000:1600:1000:1100)=(0.35,0.28,0.18,0.19). The varying computation distribution makes it challenging to determine the optimal model partition strategy. This includes the **Inter-Stage and Intra-Stage problems (over time)**.
>   - **Memory**:  The memory consumption for Stage-1 at different time points is as follows: 50 GB at T-0, 80 GB at T-1 for DP-0, 68 GB at T-0, and 60 GB at T-1 for DP-1. To prevent program crashes, we must configure the re-computation strategy based on the highest memory usage, which is 80 GB. This approach introduces additional computational overhead, categorized as the **Intra-Stage imbalance problem (over time)**. Additionally, for DP-1, the memory usage across different stages at both T-0 and T-1 has a high standard deviation (6.5 GB and 8.4 GB, respectively). Using the same re-computation strategy for different stages will also bring computational overhead according to the highest memory usage. This variation represents the **Inter-Stage imbalance problem**.
>
>
> The next part is in Part 2

---

> ### Author Response · Authors · 2024-11-19
> **Response to Reviewer dcak （2）**
>
> ## Differences between VLM and LLM training
>
>   The unique challenge in VLM training arises from its nature in model structure and data composition:
>   - **Model Structure**：  Most current LLMs are based on the Transformer architecture. The model partition method, which divides stages by parameters, ensures that stages are equally distributed. For both fixed and variable inputs, the forward time across different stages remains consistent due to the fixed structure. VLM Instruct-tuning training complicates the partition strategy due to its inherent heterogeneous model.  For fixed inputs, there is a serious computation difference between stages (inter-stage problem across devices). For non-fixed inputs, this is a huge computation distribution difference between iterations (intra-stage problem over time).
>   - **Data Composition**:  For the LLM-Pretrain task, since the input is fixed, the forward time and memory cost for the same stage remain relatively consistent across different devices and times, eliminating intra-stage imbalance problems. In the LLM-SFT task, dynamic input can be converted into static input through simple packing [4], which helps reduce computational imbalance.  In VLM instruct-tuning, however, it is hard to ensure fixed inputs for VIT and LLM by simple packing. Even if we maintain consistency in the total token count, the proportion of Vision and LLM tokens may vary. For instance, at T-0 and T-1, the total token length is 4k, but the Vision part occupies 3k tokens at T-0 and only 0.5k tokens at T-1. This discrepancy leads to significant differences in forward time.
>
> ## Motivation
>
>   Our paper focuses on the unique computational imbalance challenges encountered during VLM instruct-tuning training. These challenges differ significantly from those in LLM training, as well as from issues related to inference and serving ([1][2]). This is a complex, systematic joint optimization problem involving data, models, and memory, each influencing the others. Further clarification will be provided in the following responses.
>
> ## Response to concerns
>
> **W1:**    To quantify this problem, we used the InternVL-Chat-1.2 dataset (Section 5.1) to perform profile statistics, as shown in the following table. For the Intra-Stage analysis, we used Stage 1 as a sample for counting information.
> | Imbalance Dimension | Input Mean ± STD token (data aspect) | Time Mean ± STD ms (model aspect) | Memory Mean ± STD (G) (memory aspect) |
> |---------------------|-------------------------------------|----------------------------------|---------------------------------------|
> | Inter-Stage         | 1420 ± 955                          | 85 ± 93                          | 39 ± 23                               |
> | Intra-Stage-1       | 1975 ± 1272                         | 136 ± 155                        | 73 ± 6                                |
>
>   - **Inter-Stage**: The table highlights a significant imbalance between data, model, and memory utilization. Focusing on the model's forward time as an example, the mean is 85ms with a standard deviation of 93ms. This substantial deviation indicates considerable variability in forward time across different stages.
>   -  **Intra-Stage 1**:  The table also reveals a serious imbalance issue within the intra-stage performance. In Stage 1, the mean forward time is 136ms, with a substantial standard deviation of 155ms, indicating significant variability. Similarly, the standard deviations for length and memory are notably large relative to their respective means, further emphasizing the imbalance.
>
> **W2**： **Our contributions are reflected in three levels: framework, method, and results.**
> -  **From the framework perspective:**
>
>     We are the first to identify and address the computational imbalance problem in large-scale VLM (Vision-Language Model) instruct-tuning training, proposing a systematic solution tailored to this challenge. The computational imbalance in VLM instruct-tuning represents a complex and unique issue that cannot be directly resolved using previous LLM solutions [3] or inference approaches ([1][2]) due to the distinct demands and requirements inherent to VLM training.
>
>     Regarding the adaptive data batching method mentioned by Reviewer dcak, we find that it fails to tackle the core imbalance issue in VLM instruct-tuning training based on previous problem analysis.  Similarly, AdaPipe, as referenced by the reviewer, pertains to LLM pretraining tasks and does not align with or address the specific complexities associated with VLM instruct-tuning. This distinction was emphasized in our prior analyses and discussions, underscoring the need for a specialized, systematic approach to resolving these challenges.
>
> The next part is in Part 3

---

> ### Author Response · Authors · 2024-11-19
> **Response to Reviewer dcak （3）**
>
> -  **From the method level:**
>
>     Our approach is systematic rather than isolated and incremental, integrating the three interconnected modules of data, model, and memory to ensure computational balance across both inter-stage and intra-stage levels.
>
>     **Data**:  We are the first to identify the data imbalance problem and propose an iterative method to approximately address this challenge by keeping VIT and LLM inputs fixed simultaneously. This method is both simple and effective, processing ten million data in just tens of seconds, leading to rapid convergence in many scenarios, as illustrated in Figure 3.
>
>     **Model**: Considering the **VLM architecture and P2P (point-to-point) communication** within pipeline parallelism, we employed a novel search method. As shown in Table 5, this method outperforms traditional profile-based techniques, achieving faster performance.
>
>     **Memory**: We utilized a straightforward but highly effective greedy approach to optimize memory usage. As demonstrated in Table 6, achieving a computational balance between data and model at both inter and intra-stage levels is essential for attaining balanced memory utilization. This comprehensive integration of data, models, and memory highlights the cohesive strength of our systematic approach.
>
> -  **From the results level** :
>
>     Compared with the open-source training code of InternVL-Chat, training time is reduced greatly, achieving about 1.8x speed-up while maintaining model performance. Our method’s efficacy and generalizability are further validated across various models datasets and tasks.
>
>
> ## Common questions
>
> > **Q1:** 	The data partitioning approach in this paper appears to offer limited novelty. Numerous adaptive data batching methods, such as [1] and [2], have already addressed similar issues, albeit without specific evaluations on VLMs. However, applying these techniques to VLMs does not seem particularly challenging, given that (i) the primary goal—balancing computation both within and across batches—is quite similar, and (ii) the methodology, which involves batching based on profiled or predicted latency, closely resembles the existing works. Could the authors clarify the unique contributions of their data partitioning method in comparison to these established batching approaches?
>
> **A1:**  Our goal is to maintain consistent inputs for **both Vision and LLM** *across time (over training iterations) and across devices (both within and between devices)*, rather than merely balancing computation within and across batches. As discussed in the **Computation Imbalance Problem** and **Differences between VLM and LLM training** sections, the challenges and methods we address differ significantly from those in [1] and [2]. Numerous adaptive data batching methods discussed in [1] and [2] are specifically applicable to LLMs serving. However, while these methods support dynamic batching, they do not guarantee a fixed input across time, even less consideration was given to image and text multimodality., rendering their approaches unsuitable for our context.
>
> > **Q2:** The novelty of the model partitioning and re-computation strategies also seems limited. The AdaPipe framework, as referenced in the paper, performs layer-wise partitioning and adaptive re-computation based on computation and memory costs. Since the information needed for partitioning VLMs can also be easily profiled, could the authors explain the specific advantages of their approach over extending AdaPipe for VLMs?
>
> **A2:** Our approach at the model partition level is more comprehensive and yields better results. For instance, we consider the impact of **P2P communication** between different pipeline stages on training speed, as verified in Table 5. In contrast, AdaPipe is specifically designed for the LLM-Pretrain task and cannot be directly applied to VLM instruct-tuning training. The previous **Differences between VLM and LLM training** sections introduced the specific differences. Regarding the memory balancing strategy, our method is more straightforward compared to AdaPipe. AdaPipe employs a more complex re-computation strategy, assuming a fixed network structure and applying it to finer-grained network modules.
>
>
> The last part is in Part 4

---

> ### Author Response · Authors · 2024-11-19
> **Response to Reviewer dcak （4）**
>
> > **Q3:** The motivation for addressing data imbalance would be stronger with quantitative results from real datasets illustrating variation in sample sizes or computational demands. The simplified example in Figure 2 does not adequately demonstrate the scope of the issue.
>
> **A3:**  In Table 1, we illustrate the computation imbalance problem using the InternVL-1.2M dataset. We also introduce the DistRatio to effectively measure the degree of data imbalance. Figure 3 presents an analysis using three widely used datasets—InternVL-1.2M, LCS558K, and LLava-665K—across various image scales and training settings. Table 4 highlights how different methods impact the DistRatio, reducing it from 0.3 to around 0.1, thereby ensuring better data balance.
>
> > **Q4:** The section on memory imbalance in Section 3 lacks clarity. While it states the need for aggressive re-computation, it would benefit from a discussion on potential opportunities to mitigate memory imbalance.
>
> **A4:**  We employed a straightforward approach, as outlined in Appendix 1.1 and detailed in Table 6, which provides comprehensive step-by-step explanations and final experimental results. By adopting a greedy strategy, we minimized the amount of re-computation required at each stage. Additionally, the strategy effectively reduced the possibility of program crashes due to the constraints of the actual profile.
>
> > **Q5:** The figure in Appendix A.1.2 does not clearly illustrate how computation is balanced, making it difficult to assess the efficacy of the approach.
>
> **A5:**  Appendix A.1.2 is purposed to illustrate our training pipeline. Tables 2, 3, 4, and 5 in the body text quantitively demonstrate the effectiveness of our method in achieving computational balance across multiple datasets and models. Additionally, as shown in Figure 2, addressing the issues related to data, model, and memory helps resolve the computation imbalance problem.
>
> > **Q6&Q7:** Section 4 introduces Q'v and Q't to reduce Pad Ratio and Dist Ratio, but the paper lacks a clear formula or explanation of how these values affect the ratios. This makes it challenging to understand how Q values are chosen to achieve the intended balance.
> In Section 5, Q′v is set equal to Qv, while Q′t is defined as Qt − 128. Could the authors provide an intuition or rationale behind these choices?
>
> **A6&A7**:
> Thank you for your question. We currently do not provide a specific hyperparameter ablation experiment. Here we provide some ablation experiments for choosing  Q′v and  Q′t: Q′v represents the threshold for the number of input images, while Q′t represents the threshold for text length. It is challenging to maintain an exact number of images and text length, so we relax these conditions to allow for approximation.
>
> **A6:**  The  ablation results of Q'v and Q't are as follows:
>
> |  Q'v       |     Q’t       | DistRatio_VIT | DistRatio_LLM |
> |:------------:|:---------------:|:---------------:|:---------------:|
> |  Qv  | Qt - 64 | 0.0156        | 0.145         |
> | Qv   | Qt - 128  | 0.0159        | 0.144         |
> |  Qv   | Qt - 256  | 0.018         | 0.141         |
> | Qv - 1  |  Qt - 128   | 0.0417        | 0.20          |
>
> The above results can be obtained using the release code https://github.com/anonymousiclr293/omnibal_example/blob/main/test_balanced_dynamic_batch.py.
>
> **A7:**  Q'v represents the number of images, with each image equivalent to 1,024 tokens. This large granularity can significantly increase the DistRatio (DistRaito_VIT from 0.0159 to 0.0417; DistRatio_LLM from 0.144 to 0.20).
>
> > **Q8:** The partitioning approach begins with an anchor partition strategy. What is the computational overhead for generating this anchor partition, and how close is it to the optimal partition? If it is far from optimal, what steps are taken to improve upon it? Conversely, if it is close, is it necessary to generate multiple new candidate partitions?
>
> **A8:**  **The computational overhead is minimal**. In fact, establishing the basic anchor strategy involves profiling the forward time and activation values of different layers. In our experiment, only five steps are needed, resulting in very little computational overhead.
>
> **The anchor may not be the direct optimal solution**.  We empirically find that the final optimal solution is generally within r ≤ 3.
>
> **Cost of Calculating the Optimal Solution:** After obtaining the anchor, we select the top K partitions (10-15) near the anchor for speed measurement, as described in Section 4.2. Each speed measurement requires only 3 steps, making the cost relatively small compared to the entire training process.

---

> ### Author Response · Authors · 2024-11-22
> **Reference in Comment**
>
> [1] Yu, Gyeong-In, Joo Seong Jeong, Geon-Woo Kim, Soojeong Kim, and Byung-Gon Chun. "Orca: A distributed serving system for {Transformer-Based} generative models." In 16th USENIX Symposium on Operating Systems Design and Implementation (OSDI 22), pp. 521-538. 2022.
>
> [2] Choi, Yujeong, Yunseong Kim, and Minsoo Rhu. "Lazy batching: An sla-aware batching system for cloud machine learning inference." In 2021 IEEE International Symposium on High-Performance Computer Architecture (HPCA), pp. 493-506. IEEE, 2021.
>
> [3] Zhenbo Sun, Huanqi Cao, Yuanwei Wang, Guanyu Feng, Shengqi Chen, Haojie Wang, and Wenguang Chen. Adapipe: Optimizing pipeline parallelism with adaptive recomputation and partitioning. In Proceedings of the 29th ACM International Conference on Architectural Support for Programming Languages and Operating Systems, Volume 3, pp. 86–100, 2024
>
> [4]  Mario Michael Krell, Matej Kosec, Sergio P Perez, and Andrew Fitzgibbon. 2021. Efficient sequence packing without cross-contamination: Accelerating large language models without impacting performance. arXiv preprint arXiv:2107.02027

---

> ### Author Response · Authors · 2024-11-23
> **Reaching the End of the Public Discussion Phase**
>
> Dear Reviewer,
>
> Thank you for your time and effort in reviewing our manuscript and for providing insightful feedback. We hope our responses have adequately addressed your concerns. We welcome further discussion if you have more questions or suggestions.  With the discussion deadline approaching, we would greatly appreciate it if you could kindly take a moment to review our reply. We greatly appreciate your consideration and look forward to your thoughts.

---

> ### Author Response · Authors · 2024-12-02
> **Urgent Follow-up to Reviewer dcak – Discussion Deadline Approaching**
>
> Dear Reviewer dcak,
>
> We hope this message finds you well. We apologize for reaching out again, but with the discussion period nearing its end, we wanted to ensure that our recent updates have adequately addressed your concerns.
>
> If you have any further questions or need additional clarification, please don’t hesitate to let us know. We greatly value your feedback and appreciate the time and effort you’ve dedicated to reviewing our work.
>
> Thank you once again for your thoughtful input and continued support. We look forward to hearing from you.
>
> Best regards,
> Authors of Paper #2146

---

### Author Response · Authors · 2024-12-02
**Response to all Reviewers and Area Chairs**

Dear Reviewers and Area Chairs

We sincerely thank the reviewers and the area chair for their insightful feedback and the time they dedicated to evaluating our work. We deeply appreciate their recognition of the strengths of our research. The reviewers collectively agree that:

**The Framework:**

- **Important  Problem**  "This paper addresses a critical issue: accelerating distributed training for large-scale vision-language models (VLMs)." (**Reviewer dcak**)
- **Clear Motivation** "The motivation is clear and the in-depth analysis and ablations are convincing." (**Reviewer Fc4X**)
-  **Comprehensive Solution** "The authors present a comprehensive optimization framework that targets VLM distributed training from three perspectives: data, model, and memory balance." (**Reviewer dcak**)

**Strong Practical Contributions:**
- **Effective** "The paper centers on the efficient parallel training of MLLMs particularly focusing on balancing the computational load concerning data, model, and memory distribution both in-device and cross-device." (**Reviewer r51z**)
- **Simple but Effective** "The multi-modal sample balancing strategy stands out for its simplicity, offering a straightforward approach to load management ..." (**Reviewer r51z**)

**Comprehensive Experiments:**
-  "It is comprehensively validated across different datasets, tasks, resolutions and hardware archs etc..." (**Reviewer r51z**)
- "Extensive experiments demonstrate the approach's effectiveness for rebalancing computation for MLLMs. "(**Reviewer Fc4X**)

---

### Comment · Area_Chair_aKnA · 2024-12-03
**Discussion due soon**

Dear all reviewers,

Our reviewer-author discussion will end soon. For each of you, please check all the files and see if anything you'd like to discuss with authors.

Best,
Your AC

---

### Note · Authors · 2025-01-23

I have read and agree with the venue's withdrawal policy on behalf of myself and my co-authors.